# Image-Compression Techniques: Classical and “Region-of-Interest-Based” Approaches Presented in Recent Papers

**DOI:** 10.3390/s24030791

**Published:** 2024-01-25

**Authors:** Vlad-Ilie Ungureanu, Paul Negirla, Adrian Korodi

**Affiliations:** Automation and Applied Informatics Department, University Politehnica Timisoara, 300006 Timisoara, Romania; adrian.korodi@upt.ro

**Keywords:** region-of-interest detection, lossy and lossless compression algorithms, image-compression techniques

## Abstract

Image compression is a vital component for domains in which the computational resources are usually scarce such as automotive or telemedicine fields. Also, when discussing real-time systems, the large amount of data that must flow through the system can represent a bottleneck. Therefore, the storage of images, alongside the compression, transmission, and decompression procedures, becomes vital. In recent years, many compression techniques that only preserve the quality of the region of interest of an image have been developed, the other parts being either discarded or compressed with major quality loss. This paper proposes a study of relevant papers from the last decade which are focused on the selection of a region of interest of an image and on the compression techniques that can be applied to that area. To better highlight the novelty of the hybrid methods, classical state-of-the-art approaches are also analyzed. The current work will provide an overview of classical and hybrid compression methods alongside a categorization based on compression ratio and other quality factors such as mean-square error and peak signal-to-noise ratio, structural similarity index measure, and so on. This overview can help researchers to develop a better idea of what compression algorithms are used in certain domains and to find out if the presented performance parameters are of interest for the intended purpose.

## 1. Introduction

Image compression is a vital technique with numerous applications at the moment. In our increasingly digital world, images play a pivotal role in communication, information sharing, and artistic expression. However, the importance of efficient image compression cannot be overstated. This represents a technology that allows users to reduce the file size of images while preserving an acceptable level of visual quality. This process has far-reaching implications for a multitude of industries and everyday digital interactions, making it an integral part of the current digital landscape. One of the primary reasons why performant image compression is crucial is its ability to reduce the storage space required for image files. In the era of high-resolution images and digital photography, image files can quickly become unwieldy. Compression significantly decreases the size of these files, making them more manageable and economical in terms of storage. Whether it is personal photo collections, business databases, medical information, or scientific archives, the efficient use of storage resources is essential. The importance of image compression also extends to the realm of data transmission making it possible for images to be transmitted more swiftly over networks or buses, which is essential in various applications. In the age of e-commerce, for example, faster-loading images on websites can significantly improve the user experience and increase sales. Video streaming platforms also rely on image compression to deliver content seamlessly, ensuring uninterrupted playback. Faster transmission is mandatory for automotive or medical sectors [1] where bandwidth efficiency is vital. 

Image compression not only accelerates data transmission but also optimizes bandwidth usage. This is especially important in areas with limited or expensive bandwidth, such as in an autonomous car driving system, ensuring that users can access image-rich content without excessive data consumption. In regions where internet access is a precious resource, image compression can make a substantial difference in accessibility. Another big field where image compression is vital is the medical field, where magnetic resonance imaging (MRI), ultrasound or computed tomography (CT) images must be stored, transmitted, and analyzed by doctors. In this case, the clarity of the details is of utmost importance, and a poorly compressed image could result in an erroneous diagnosis [2]. For businesses and individuals, image compression can translate into significant cost savings, by reducing storage and bandwidth requirements, expenses related to infrastructure, cloud storage, and data plans are minimized. These savings can be particularly meaningful for companies dealing with vast amounts of image data or for individuals managing their personal media collections.

Modern image-compression techniques strive to strike a balance between reducing file size and maintaining image quality [3]. In applications where image fidelity is paramount, such as medical imaging, art preservation, and professional photography, image compression allows for the preservation of critical visual details while optimizing storage and transmission. In the end, efficiency in digital operations extends to energy consumption; compressed images reduce the energy required for devices to render or transmit images. This is especially crucial for battery-powered devices and mobile technologies, where energy conservation can extend battery life and reduce the environmental impact. In real-time applications like video conferencing, gaming, advanced-driver-assistance systems (ADAS), telemedicine, and any type of data live streaming, fast and efficient image compression is essential for maintaining low latency and consistent performances.

Image compression techniques include lossless and lossy methods [4], with the choice between them depending on the specific requirements of the application. The method used for data compression and the quality of the compressed file that results are the primary distinctions between lossy and lossless compression. Lower-quality files are produced using lossy compression, which compromises data integrity and clarity to reduce file size. Lossless compression maintains data integrity for higher-quality files and is frequently employed for text-based files when data integrity is crucial, but, in some cases, the file-size reduction might not be as significant. 

By removing some of the data from the image file, lossy image compression shrinks the file size overall. Because of the irreversible nature of this process, the file information will be permanently deleted. Discrete wavelets transform, fractal compression, and transform encryption are a few of the algorithms utilized in lossy compression. These techniques can significantly lower the file size, but doing so typically comes at a cost meaning that the image quality will suffer. Having a backup file before making any changes is, therefore, preferable. A great illustration of lossy compression is the JPEG file format which works well for images and photos that lack transparency.

Lossless image compression is designed to preserve image quality by efficiently encoding and storing all original data. This process eliminates redundant information but ensures that the image can be almost perfectly reconstructed from the compressed file. The drawback of this method is that the reduction in file size is often not as significant as with lossy compression, which may limit its usefulness in saving storage space. Run-length encoding, arithmetic encoding, and Huffman coding are examples of common lossless compression algorithms. The ideal photographs for the lossless compression approach are those with a transparent background and a lot of text. The file formats like RAW, BMP, GIF, and PNG can all benefit from lossless-image-compression algorithms. To better understand the difference in file size, for lossy compression a size reduction of 85% can be achieved for an image, while, using a lossless compression, for the same image only a 5% reduction is recorded. Both methods have their advantages and disadvantages, depending on the use case the user shall decide what approach is suitable.

In domains such as medical-image processing, the large size of images, the need for efficient storage, and the crucial requirement of preserving the details and the clarity of the images (so the experts can provide an accurate diagnosis based on them) created the need for a hybrid compression method [5]. Therefore, two areas have been defined, region of interest (ROI), and region of not interest (RONI or non-ROI). Important diagnostic information is contained in medical photographs, which must be retained during compression. The classical compression approaches apply the same lossless compression algorithm to the whole image to maintain each component of a medical image. Preserving diagnostic features is the reason for using lossless compression technology on medical photographs. However, because the regions of high importance are treated with the same priority and in the same manner as the background of the image, the resulting compression ratio (CR) will be low and the transmission of the image on the web requires a high bandwidth [6]. The ROI-based compression technique can be used to alleviate this issue [7].

The term “region of interest” (ROI) refers to the area of a picture that is more significant than other areas [8], while non-ROI refers to the rest of the image. Applying lossy compression on non-ROI and lossless compression on ROI leads to maintaining the necessary diagnostic features of the medical image while achieving a high compression ratio. Utilizing ROI-based compression has the benefit of offering a high compression ratio without compromising the quality of a significant portion of the image [9]. Because an image’s compression ratio is strongly related to its ROI size (a smaller ROI means better CR), precisely defining the size and location of the diagnostic zone from medical pictures becomes very important. It is crucial to appropriately detect ROI, since it includes crucial diagnostic information that cannot be altered, and the CR of the image is dependent on its size. The detection of ROI is a vital task since incorrectly located ROI might result in the loss of diagnostic data in medical pictures. One of the first steps in any ROI-based compression technique is ROI selection. The complexity and execution duration of any approach are determined by how the ROI is chosen. Some common ROI-selection strategies (such as region growth and saliency maps) are employed frequently by different researchers, or sometimes the ROI is chosen manually if the images follow a certain pattern. After the regions are defined, different compression techniques are applied to preserve the region of interest and to save space by sacrificing the non-ROI. Usually, to reconstruct the input image, the inverted process is applied during the decompression step. Figure 1 shows the general process of ROI-based compression and highlights the main steps that are followed to achieve hybrid image compression. Of course, that extra steps such as filtering, salt and pepper noise reduction or image watermarking (used for authentication and tampering prevention) are to be expected.

This paper aims to present the latest ROI-based approaches showing the differences in the utilized principles and algorithms from the existing classical methods. Commonly used metrics are shown to appreciate the quality and clarity of the images subjected to the compression and decompression process. This study is oriented towards the individual compression of the ROI areas for applications in the automotive field (autonomous driving, car traffic-safety systems, systems incorporated in vehicles to help drivers) and the medical or telemedicine fields (magnetic resonance imaging, X-ray, computer tomography, ultrasound imaging). Depending on the purpose of the applications where the images are used, ROI selection and compression can be treated as separate issues, but both topics always require a specific analysis.

## 2. Relevant Performance Metrics

The compression approaches compute their efficiency and performance using a variety of performance criteria.

The mean-square error (MSE) is defined as the description of the cumulative squared error between the compressed image and the original image [10], and it gives perspective of the difference between these images. A lower value of MSE means a more efficient compression algorithm. The equation used to define MSE is presented in (1):(1)$$MSE=\frac{1}{M\times N}\sum_{j=1}^{M}\sum_{i=1}^{N}{\left(X_{j,i}-{X^{\prime }}_{j,i}\right)}^{2}$$
where M and N represent the size of the image, X the actual values, and X′ the expected values. 

Peak signal-to-noise ratio (PSNR) is defined as the ratio of the maximum pixel intensity to the mean-square error [11], and it is measured in decibels (dB). A higher value for the PSNR metric represents a better image quality, and it is typically used to test the considerable distortion introduced by the compression and decompression process between the input and output image [12]. PSNR is presented in (2) and is computed based on MSE:(2)$$PSNR=10\log_{10}\left(\frac{{\left(2^{B}-1\right)}^{2}}{MSE}\right)$$
where B is the number of bits per pixel; for example, if pixels are represented using 8 bits per sample, then B will also be 8.

Compression ratio (CR) is defined as the ratio of the size (usually expressed in multiples of bytes) of the original image to the size of the compressed image. A higher number means a better compression ratio which can facilitate the storage and transmission of an image but also can lead to poor image quality. The CR is computed using (3):(3)$$CR=\frac{Size\;of\;the\;original\;image}{Size\;of\;the\;compressed\;image}$$

Bits per pixel (BPP) is defined as the ratio of the total size of the compressed image to the total number of pixels in the image [13]. It can be computed based on (4). Bits per pixel express the average number of bits required to represent an image’s pixel information. The size of an image depends on its width (in pixels), height (in pixels), and the number of bits per pixel.
(4)$$BPP=\frac{Size\;of\;the\;compressed\;image}{Total\;number\;of\;pixels\;in\;the\;image}$$

Structure similarity index (SSIM) is used to measure the tendency for similarity between the original image and the compressed image [14]. SSIM is based on visible structures inside the image and can predict the perceived quality of the compressed image. The resulting value of this metric is between −1 and 1, when discussing identical images, the output of SSIM will be 1. The formula used to find SSIM for two windows x and y of common size N × N is defined in (5) [15]:(5)$$SSIM\left(x,y\right)=\frac{\left(2\mu_{x}\mu_{y}+c_{1}\right)\left(2\sigma_{xy}+c_{2}\right)}{\left(\mu_{x}^{2}+\mu_{y}^{2}+c_{1}\right)\left(\sigma_{x}^{2}+\sigma_{y}^{2}+c_{2}\right)}\;\left\{\begin{array}{c} c_{1}={\left(k_{1}L\right)}^{2},\;k_{1}=0.01\\ c_{2}={\left(k_{2}L\right)}^{2},k_{2}=0.03\\ L=2^{BPP}-1 \end{array}\right.$$
where $\mu_{x}$ is the pixel-sample mean of window x; $\mu_{y}$ is the pixel-sample mean of window y; $\sigma_{x}^{2}$ is the variance of window x; $\sigma_{y}^{2}$ is the variance of window y; $\sigma_{xy}$ is the covariance of the two windows; $c_{1}$ and $c_{2}$ are two variables to stabilize the division with a weak denominator; and L is the dynamic range of the pixel values.

Signal-to-noise ratio (SNR) can be defined as the ratio of the signal power to the noise power. It is measured in dB and can be computed as stated in (6):(6)$$SNR=10\log_{10}\left(\frac{P_{signal}}{P_{noise}}\right)$$
whare $P_{signal}$ is calculated as the mean of the pixel values, and $P_{noise}$ is calculated as the standard deviation or error value of the pixel values.

Percentage rate of distortion (PRD) is defined as the measure of the distortion in the reconstructed image. The lesser the value of the PRD in the reconstructed image the less it is distorted [16]. PRD for an image with the size M × N can be computed using (7): (7)$$PRD=100\sqrt{\frac{\sum_{j=1}^{M}\sum_{i=1}^{N}{\left(X_{j,i}-{X^{\prime }}_{j,i}\right)}^{2}}{\sum_{j=1}^{M}\sum_{i=1}^{N}{\left(X_{j,i}\right)}^{2}}}$$
where X represents the original image, and X′ represents the reconstructed image.

Computational time (also called running time RT) is defined as the amount of time required in the process of compression and decompression of an image. An efficient technique must have less computational time. The RT is usually obtained using different time-measurement tools and techniques. 

Mean opinion score (MOS) is defined as the visual assessment of reconstructed image quality by comparing it with the original image. The ranking is conducted with the help of an integer scale from 1 to 5, where 1 is lowest perceived quality, and 5 is the highest perceived quality. The MOS metric is calculated as the arithmetic mean over single ratings performed by human subjects for a given stimulus in a subjective quality-evaluation test, the equation used is described in (8):(8)$$MOS=\frac{\sum_{n=1}^{N}R_{n}}{N}$$
where N is the number of subjects, and $R_{n}$ is the individual rating of subject n.

Structural content (SC) is used to provide a comparison between two images inherited in small patches and to determine the common parts that images have [10]. The higher the value of SC, the poorer the quality of the image. The SC is defined using (9):(9)$$SC=\frac{\sum_{j=1}^{M}\sum_{i=1}^{N}{\left(X_{j,i}\right)}^{2}}{\sum_{j=1}^{M}\sum_{i=1}^{N}{\left({X^{\prime }}_{j,i}\right)}^{2}}$$
where M and N represent the size of the image, X represents the original image, and X′ represents the reconstructed image.

The correlation coefficient (CC or CoC) is used to describe the existing correlation between the original image and the reconstructed image [17]. The correlation coefficient is between −1 and 1. An absolute value of 1 indicates that a linear equation properly represents the relationship between X and Y, with all data points resting on a line; this can be reached only for identical images. The regression slope determines the sign of the correlation: a value of +1 suggests that all data points lie on a line where Y rises as X increases, and vice versa for −1. A value of 0 indicates that the variables are not linearly related. The equation used to compute CC is presented in (10):(10)$$CC=\frac{n\sum x_{i}y_{i}-\sum x_{i}\sum y_{i}}{\sqrt{n\sum x_{i}^{2}-{\left(\sum x_{i}\right)}^{2}}\sqrt{n\sum y_{i}^{2}-{\left(\sum y_{i}\right)}^{2}}}$$
where n is the sample size, $x_{i}$ and $y_{i}$ are individual sample points indexed with i.

Sometimes, instead of the compression ratio (CR), the space-saving (SS) metric can be used. Space saving is defined as the reduction in size relative to the uncompressed size and sometimes can be noted as a percentage. The equation used to compute the SS value is depicted in (11):(11)$$SS=1-\frac{Size\;of\;the\;compressed\;image}{Size\;of\;the\;original\;image}$$

The visual saliency-based index (VSI) [18] is a full-reference image-quality-assessment (IQA) index that measures the perceptual quality of an image. It is based on the concept of visual saliency (VS), which refers to the areas of an image that attract the most attention of the human visual system. The formula for VSI is not straightforward and involves several steps. The VSI model uses visual saliency as a feature when computing the local quality map of the distorted image. Additionally, when pooling the quality score, VS is employed as a weighting function to reflect the importance of a local region. VSI is defined using Equation (12):(12)$$VSI=\frac{\sum_{x\in \Omega }S\left(x\right)\times VS_{m}(x)}{\sum_{x\in \Omega }VS_{m}(x)}$$
where x is a given position where the similarity shall be measured, Ω is the whole spatial domain, S(x) is the local similarity at each location x, and $VS_{m}(x)$ is used to weigh the importance of S(x) in the overall similarity.

The number-of-pixels-changed ratio (NPCR), which is precisely computed by comparing the pixels of the initial images and watermarked image, generally indicates the number of pixels altered by the integration process. If the two images that are compared have M × N pixels in size, the following Formula (13) provides a precise definition of the NPCR:(13)$$NPCR=\sum_{j}^{M}\sum_{i}^{N}\frac{D(i,j)}{T}\times 100\%\;D(i,j)=\left\{\begin{array}{c} 0,\;\mathrm{if}\;P_{1}\left(i,j\right)=P_{2}(i,\;j)\\ 1,\;\mathrm{if}\;P_{2}\left(i,j\right)\neq P_{2}(i,\;j) \end{array}\right.$$
where T is the size of the image and $P_{1}\left(i,j\right)$ and $P_{2}\left(i,j\right)$ are the pixels from the original and watermarked image.

The root mean square error (RMSE) is a commonly used metric for comparing numbers (population values and samples) that are predicted using an estimator or a model. The sample standard deviation of the variations between the expected and observed values is described using this metric. When the calculations are performed over the data sample that was used for estimation, each of these differences is referred to as residual. When they are performed outside of the sample, they are referred to as prediction errors. The RMSE creates a single indicator of predictive power by combining the magnitudes of the errors made in predicting various times. A lower value indicates a better match between two images. The equation used to define RMSE is presented in (14):(14)$$RMSE=\frac{1}{MN}\sqrt{\sum_{j}^{M}\sum_{i}^{N}\left(P_{1}\left(i,j\right)-P_{2}\left(i,j\right)\right)}$$
where M × N is the size of the image, and $P_{1}\left(i,j\right)$ and $P_{2}\left(i,j\right)$ are the pixels from the original and the output image.

The mean absolute error (MAE) [19] is a statistical measure of inaccuracies between paired observations that represent the same phenomenon. Comparisons of predicted versus observed data, subsequent time versus initial time, and one measurement technique versus another are a few examples of Y versus X. The MAE is computed by dividing the total absolute errors by the sample size. A lower value indicates a better match between two images. The equation used to define MAE is presented in (15):(15)$$MAE=\frac{1}{MN}\sum_{j}^{M}\sum_{i}^{N}\left|\left(P_{1}\left(i,j\right)-P_{2}\left(i,j\right)\right)\right|$$
where M × N is the size of the image and $P_{1}\left(i,j\right)$ and $P_{2}\left(i,j\right)$ are the pixels from the original and the output image.

Other metrics used to analyze the quality of images and that can be mentioned are average difference (AD), maximum difference (MD), normalized cross-correlation (NK), image fidelity (IF), normalized absolute error (NAE), distortion index (DI) or mean-structural-similarity index (MSSIM).

## 3. Classical Image Compression Approaches

To develop a better overview of the actual benefits of ROI-based methods, it is worth having an overview of the latest advances of other lossless and lossy compression techniques. By comparing different state-of-the-art classical approaches with the hybrid methods that will be presented in Section 4, the computational costs and complexity associated with the process of defining a region of interest can be justified or not. Therefore, this chapter will present different compression techniques that handle the whole picture in the same way. Meaning that the algorithms are not applied on some portions of interest. Relevant metrics will be considered when discussing about the image quality and compression performances. The chapter will be divided in subchapters that will emphasize the domain in which the described algorithms were applied.

### 3.1. Methods Used in Medical Imaging and Telemedicine

This is one of the main areas in which image compression is used in the medical field. Focusing on this domain, Dokur et al. [20] propose a framework for image compression and segmentation using artificial neural networks. The author uses two neural networks, a Kohonen map and incremental self-organizing map (ISOM). First, the input image is separated in 8 × 8 pixels blocks, and then, two-dimensional discrete cosine transform coefficients are calculated for each resulting block. A low-pass filter is applied to eliminate high frequencies that are not visible to the human eye. The output of the low-pass filter, codeword, serves as an input for the learning process of the neural network. Computer simulations were performed, and different quality metrics (CR and MSE) were computed. It is observable from the provided metrics that a Kohonen map does not provide a satisfactory output. Even if the compression ratio is better than for the JPEG method (31 compared to 22.94), the MES has a bigger value of 139.91 compared to 102.32 for JPEG. For sure, even if the incipient results were not great, the whole-image compression based on a neural network approach shall not be excluded for future work in this domain.

Fast and safe transmission of images is vital in the medical domain; this is why different compression methods must be applied. The lossy compression type can achieve high compression ratios, but the image quality is affected, while the lossless compression type retains the image quality but does not have a good compression ratio. Starting from these two types of compression, the authors of paper [21] (Punitha et al.) presents the idea of a near-lossless compression which can achieve satisfactory results for both compression rate and image quality. The proposed method uses sub-bands thresholds to increase the number of zero coefficients. Entropy run-length encoding is used to compress the image to retain valuable information and quality. The first step of the proposed method consists of wavelet decomposition completed using discrete wavelet transform using db1 wavelet. Then, the separation of coefficients is performed, and four sub-bands are obtained using approximation and detailed coefficients. The next step is called thresholding coefficients, which is a process used to isolate the relevant information from the image. Encoding the sub-bands using RLE is completed next. RLE is a lossless encoding mechanism used to compress images. The final part is the decompression process where the sub-bands are decoded, and the image is reconstructed. Metrics such as CR, PSNR, time complexity and BPP were used to examine the results. The achieved performances were compared to other consecrated and generally used compression techniques. For the evaluation of 256 × 256 images with 8 bit depth were used. The average CR was 5.67, and the average PSNR was 42.43 dB, meaning that from a visual point of view the output image can be considered almost identical to the original one. The shortcoming of the utilized method was that the image-size reduction is not significant.

Vallathan et al. in paper [22] focus on a lossless image compression that relies on hierarchical extrapolation using Haar transform and the embedded encoding technique. First, the color image is decomposed to Y, $C_{0}$, and $C_{g}$ by applying color transform depicted in (16). The luminance component Y is processed using Haar transformation and then using SPIHT encoding [23]. When encoding the chrominance component, a hierarchical extrapolation scheme is used and then Huffman coding is applied. For the decompression part, SPIHT and Huffman decoding are applied, and the image is reconstructed. The utilized compression techniques show potential when analyzed using different relevant metrics such as PSNR, CR, or MSE. The results are compared to JPEG2000.
(16)$$\left\{\begin{array}{c} Y=0.2989\times R+0.5866\times G+0.1145\times B\\ C_{0}=-0.1687\times R-0.3312\times G+0.5\times B\\ C_{g}=0.5\times R-0.4183\times G-0.0816\times B \end{array}\right.$$
where R is the red component, G is the green component, and B is the blue component of a pixel.

During compression, images are susceptible to data loss and interferences; therefore, Nemirovsky-Rotman et al. [24] proposed a compression and speckle denoising method. The algorithm is based on optimizing the quantization coefficients when applying wavelet representation on the input image. The noise reduction is an important part of the proposed method, with the objective being to reduce the effective speckle while preserving the edges in the image. Experiments and measurements were made to prove that the algorithm can simultaneously compress and despeckle the images based on the optimization of the quantization coefficients, taking into consideration an a priori version of the image. Different metrics were applied to demonstrate the value of the proposed method, and a comparison to JPED2000 was also discussed with the average PSNR metric having the value of 33.7 dB with a CR of around 25. Thes result show that the size reduction can be satisfactory, but the image quality may be too degraded for certain uses.

The storage of images for long periods of time is a necessity nowadays, but it also is a rising challenge, since there is limited storage capacity and any increase in this capacity results in extra costs. The presented study by Ammah et al. [25] propose a solution to the storage issue, a DWT-VQ (Discrete Wavelet Transform–Vector Quantization) method to compress images but at the same time to preserve its quality. The technique also aims to reduce the speckle and salt and paper noises while preserving the edges of the input image. DWT filtering is applied to the images, and then, a threshold approach is used to generate coefficients. The obtained result is then vector quantized, and Huffman encoding is applied. The best results were achieved when two levels of DWT were applied before the quaternization process. When the image is needed, the reconstruction process is applied using the Huffman decoder, VQ decoder, and inverse DWT. The proposed solution was compared to different existing methods in the literature using PSNR, SSIM, CR, or MSE metrics. The results achieved are promising; the values using PSNR metrics are around 43 dB, and for CR they are around 90.

Shahhoseini et al. in paper [26] use a lossy technique designed on wavelet transform to compress images and to preserve clinical information. The input image is converted to odd–even images in the spatial domain, therefore, resulting in four images (odd–even, odd–odd, even–even, and even–odd). Then, the wavelet transform is applied to each sub-image. Finally, an entropy encoding step is performed by applying the Huffman encoding scheme. In this manner, a compressed image is obtained; for decompression, the inverse process is executed. The obtained CR is up to 15, while the PSNR value is up to 27.8 dB.

Janet et al. in paper [27] present a joint-image lossless compression scheme based on contourlet transform. Contourlet transform is an extension of the known wavelet transform that comprises two blocks, a Laplacian pyramid (LP), and a directional filter bank (DFB). First the input image is converted to gray scale, then a two-dimensional Contourlet transform is applied. Then, global-level thresholding based on Otsu’s method is applied followed by Huffman encoding. Most image-processing algorithms produce different results when applied to different classes of medical images. The presented algorithm was used on a variety of medical images that were gathered from an MRI and CT modality database. The contourlet transform makes it possible to reconstruct images more accurately. Information-preserving image transfer is necessary for telemedical applications; thus, lossless techniques are used. The suggested system is purely lossless, and the experimental findings show that its compression percentages are better than other methods presented in the literature. The average compress ratio obtained is 14.18, and the average PSNR value is 34.44 dB.

One clinical area where scans are usually required is oral medicine; in certain situations, stomatologists use dental images to better plan an intervention. As in other situations mentioned before, the efficient storage (achieving a high CR) and transmission of information contained in images is an issue. Because dental images are analyzed by specialists, it is critical to retain useful diagnostic information (for example, having a high PSNR value) while avoiding the appearance of visible distortions caused by the applied compression technique. Starting from an image’s discrete cosine transform (DCT) and partition scheme optimization, the authors of paper [28] (Krivenko et al.) developed a noniterative approach to achieve lossless compression of dental images. In this paper, the authors examined and applied the relationships between three quality metrics: PSNR, PSNR using Human Visual System and Masking (PSNR-HVS-M), and feature similarity (FSIM) and the quantization step (QS), which controls the compression ratio. The distortion-visibility-threshold values for these metrics have been considered, while making sure that any detectable changes in noise intensity have been incorporated into the QS configuration. To achieve this, an advanced DTC coder available online was used. Tests were performed on 12 dental images with the size of 512 × 512 pixels. When testing, the observed and expected behavior was that the CR and PSNR metrics were affected by the value of the quantization step in a monotonous manner. This statement means that an increase in the QS value to a higher compression ratio was registered, but the PSNR value was dropping. Depending on the input image, the results may vary drastically, since for the same QS of 10, the CR results were inside the [6,22] interval. While for a QS of 20, the CR obtained is between 12 and 78, with a PSNR ranging from 37.5 dB to 52.5 dB. This means that the wrong selection of QS might lead to unexpected results and loss of data. Therefore, testing, and extra analyses are necessary to define a set of properties that images must have to achieve predictable results for a given quantization step. Reaching the right value means that a visual-based comparison made by an expert between the original and compressed image will lead to the conclusion that the introduced distortions are undetectable.

When talking about the software-based ultrasound-imaging systems, the high data rate required for data transfer is a major challenge, especially when referring to real-time imaging. In the current paper [29], a Binary cLuster (BL) code which yields a good compression is used by Kim et al. The starting point of the proposed method is the conventional exponential Golomb code. The aim is to compress any integers in real time without any overhead data when performing encoding and decoding; the specific steps for each operation are presented. An improved compression ratio was recorded by reducing the prefix size; the recorded values were around 40%, and in any tested scenario the CR was better than when the Golomb code was used. The average time for encoding took 0.15 s more than the Golomb code, while the average decoding time took 0.76 s less. Therefore, the average time for the whole encoding and decoding process was reduced by 0.61 s.

Certain real-time services require a very fast transfer of images to a host, and the high transfer rate required can be an issue. In the presented study [30], Cheng et al. focus on the use of MPEG technology to increase the image-compression rate for ultrasound RF data. The images were encoded and decoded using an MPEG freeware. The constant rate factor (CRF) was used as rate control for the encoding process. A high value for CRF means larger quaternization which will result in encoding the frames into a smaller size. Setting the CRF to 0 means there will be lossless compression. The aim was to reduce the compression ratio for real-time applications. The best CR recorded was 7.7.

Focusing on real-time medical processes, an algorithm for lossless compression of ultrasound sensor data is proposed in paper [31] by Mansour et al. The method is designed to operate on the transducer’s RF data after the ADC (Analog to Digital Converter) and before the receiver beamformer. The algorithm significantly reduced the data throughput by exploiting the existing redundancy in the transducer’s data. Lossless compression of the data would reduce the costs and would simplify the interfaces used for digital signal processing. The idea is to exploit the high correlation between the data of adjacent transducers to significantly reduce the energy on each line and encode the residual data instead of the original high-energy data. The compression ratios obtained are up to 3.59.

### 3.2. Methods Used for Spatial Images and Remote-Sensing Images

Because of hackers’ rising skills and the large volume of sensitive data circulating through digital channels, protecting remote-sensing photos during data transfer is crucial. Joint picture compression and encryption approaches for data transfer serve the purpose of achieving high transmission reliability, while maintaining reduced implementation costs. Existing solutions for multiband remote-sensing pictures usually have drawbacks such as long preprocessing durations, not supporting images with a high number of bands, and lack of security. To address the issues, Cao et al. [32] propose a multiband remote-sensing image joint encryption and compression algorithm (JECA), which includes several stages: preprocessing encryption phase, crypto compression phase, and decoding phase. The preprocessing stage serves the purpose of obtaining a greyscale image which is obtained by receiving and combining all the bands. The greyscale image is divided into blocks on which discrete cosine transform (DTC) is applied. After applying DTC, two coefficients called DC (which represents the low-frequency component inside the block; it is usually a large value) and AC (which represents the high-frequency component inside the block; it is usually a small value) are obtained and then encrypted in the second stage. In the final stage, the reverse processes are performed to achieve the greyscale image again. The DC and AC coefficients are decrypted, and then the blocks are restored and merged into a grayscale image. Finally, postprocessing is applied to the grayscale image to create a remote-sensing image. According to the experimental results, which were performed on a multispectral image dataset containing images with 14 bands, JECA can reduce the sender’s preprocessing time by half when compared to already-existing joint encryption and compression methods that rely on the encryption-then-compression principle. JECA also improves security while retaining the same compression ratio as existing methods, particularly in terms of visual security and key sensitivity. JECA can achieve compression efficiency comparable to JPEG and other similar algorithms. Another observed fact was that increasing the number of bands does not affect performance. For the reconstructed image, the SSIM metric was close to 0.93, and it was measured that the remote-sensing image-file size was reduced to about 5% of the original, meaning a CR of 20. The obtained PSNR values are very low, meaning that the encrypted images have a high degree of distortion. Unfortunately, the PSNR values for the reconstructed images were not available.

### 3.3. Methods Used in Automotive

Until now, a vehicle did not rely on images when making any decisions; distance sensors such as ultrasound, LIDAR, or radar were widely used in obstacle detection and avoidance features. In the context of autonomous driving vehicles, cameras are a very important part for the perception mechanism. Transmission of large images can represent a major challenge and a bottleneck for certain memories and processors. Therefore, modern encoder/decoder mechanisms are required for data transfer inside the automobile. A key aspect being computational time, because the process of encode, transmit, and decode shall be faster than just directly transmitting the image. Löhdefink et al. in paper [33] present an approach based on generative adversarial networks (GANs), with compression and decompression being made by means of an autoencoder. The authors investigated several scenarios, and, of course, when a compression was performed with a higher bit rate, better results were achieved when analyzing the metrics. An interesting result was that the semantic segmentation in a low bit rate regime yielded better results than JPEC2000 even if the PSNR metric is better for the latter method. Therefore, the method has the potential to yield high semantic segmentation performances even if the reconstruction of the original image is not as good as other conventional methods, and the PSNR metric has the value of 27.73 dB. 

### 3.4. Methods Not Targeting a Specific Domain

Several papers do not focus on a certain domain, aiming for a more general area of use when testing the algorithms; the authors usually use standard test images. For example, in paper [34], Wei et al. propose a multi-image compression–encryption algorithm based on two-dimensional compressed sensing (2D CS) and optical encryption to achieve large-capacity, fast, and secure image transmission. First, the study designs a new structured measurement matrix and applies compressed sensing to compress and encrypt multiple images at once. The multiple images are then encrypted for secondary encryption using double-random-phase encoding, which is based on the multi-parameter fractional quaternion Fourier transform. This enhances the security performance of the images. Additionally, a fractional-order chaotic system built for encryption and image compression exhibits more intricate chaotic behavior. The utilized algorithms for image encryption based on 2D CS and image compression or decompression are presented in the paper and are backed using flowcharts. The algorithm exhibits strong security and robustness, according to the experimental results. The test images used include 512 × 512 well-known color images (Lena, Peppers, Lake, and Airplane) and 256 × 256 grayscale images (Lena and Cameraman). The average CR is two, and the average PSNR value is 35.59 dB.

These days, compressing encrypted images effectively is still very difficult. In the paper [35], Wang et al. proposed a novel encryption-then-compression (ETC) scheme using heuristic optimization of bitplane allocation to improve the performance of lossy compression on encrypted gray images. To be more precise, a bitplane was used as the fundamental compression unit when compressing an encrypted image. Then, the lossy compression task was formulated as an optimization problem that maximizes the peak signal-to-noise ratio (PSNR) while adhering to a predetermined compression ratio. Asymmetric properties of various bitplanes were then utilized to create a heuristic strategy of bitplane allocation to roughly solve this optimization problem. Specifically, there are four sub-images within an encrypted image. A single sub-image is set aside, and the most important bitplanes (MSBs) of the remaining sub-images are chosen one after the other until a predetermined compression ratio is reached. The next step was to employ the low-density parity-check (LDPC) code to compress these bitplanes in accordance with the ETC framework, since there are evident statistical correlations both within and between neighboring bitplanes, where bitplane refers to those that belong to the first three MSBs. To reconstruct the original image, the authors proposed a combination of LDPC decoding, decryption, and Markov random field (MRF) exploitations to recover the selected bitplanes belonging to the first three MSBs in a lossless manner, and then they used content-adaptive interpolation to recover missing bitplanes and thus discarded pixels which are symmetric to the encrypted-image compression process. The suggested approach is feasible and effective, as demonstrated by the experimental simulation results, which also significantly outperform state-of-the-art ETC methods in achieving the desirable visual quality of reconstructed images. In the simulation, eight 512 × 512 gray images of different textures and edges (Baboon, Barb, Boat, Hill, Lena, Man, Peppers, and Tank) were tested. For a CR of two, the achieved PSNR is 38.36 dB, while for the same picture for a CR of five, the PSNR value drops to 30 dB.

## 4. Hybrid Image-Compression Approaches

Classical lossy and lossless methods take the whole input image and compress it. Only parts of that image contain useful information; therefore, in recent years, the contextual or region-of-interest approach gained more visibility. The principles are simple; from the whole image, only a region is isolated and considered of high interest. There are multiple methods of how to define the ROI, and it either can be defined manually or automatically. In this chapter, different ROI-selection methods are presented: mathematical, masked-based, segmentation, interactive, growth, or based on neuronal networks. After all, the selection of ROI is very important because metrics such as compression ratio and computational time depend on its size. After ROI and RONI parts are defined, it is a matter of choosing the right filtering and compression methods to preserve the details inside the important parts and save space by sacrificing the quality of the non-important area.

The current chapter will present the state-of-the-art methods from the last decade when discussing ROI-based compression and decompression. The general approach of each method will be discussed alongside some relevant quality metrics. The obtained results are also compared to existing methods which can be considered proper references. The chapter will be divided into subchapters that will emphasize the domain in which the described algorithms were applied.

### 4.1. Methods Used in Medical Imaging and Telemedicine

The leading field in which hybrid image compression is used is medical imaging. Hosseini et al., in paper [36], acknowledges the existing difficulties regarding the storage and transmission of medical images while still mentioning the high quality. To overcome these challenges, the authors propose a context-based method called contextual vector quantization (CVQ). This means that a region is defined as the most important part, and it shall not suffer considerable quality loss. Therefore, two versions of the CVQ algorithm are used, one for the region of interest (where a low compression ratio and high bit rate are used) and another for the background (where a high compression ratio and low bit rate are used). The two regions are separately encoded, and to reconstruct the image, they are decoded and then merged. The method proposed is like other contextual methods like CSPIHT, but it uses vector quantization instead of set partitioning in hierarchical trees. The main idea consists of the separation of the contextual region of interest and the background using growing region segmentation techniques. Different compression rates are applied for each part. The growing segmentation technique consists of choosing a point as the point of interest and, from there, growing the region of interest by appending the nearest pixels that have the same attributes. In this paper, the contextual region of interest (CROI) was extended based on the difference between the pixel’s intensity and the mean of other pixels from the same row. The obtained results were analyzed using CR, MSE, PSNR, and CoC metrics and were also compared to general methods such as JPEG, JPEG2K, and SPIHT, alongside ROI-based methods such as Maxshift, EBCOT, and CSPIHT. The presented experiments prove that even if a combined compression ratio of 256 was used, the quality of the CROI will still be very good, with the critical information still being visible and clear. The average CoC for the region-of-interest area was 0.998 while the average PSNR value was 40.16 decibels.

The whole idea behind the ROI-based techniques can be graphically summarized in the example presented in Figure 2.

A scheme for context-based encoding and decoding is presented by Ansari and Anand [37] with the goal of achieving better compression rates for an image, while maintaining the quality of the region of interest. Therefore, the contextual region of interest is defined as the area containing the most vital information that must be preserved. Following this approach, the image is split into two parts, one is the ROI which is encoded using a very low compression ratio, and the other part is the background which is encoded with a high compression ratio. The ROI is identified using segmentation and interactive methods, using a generated mask which describes the characteristics of the areas that must be encoded with higher quality. The compression method is based on contextual set partitioning in hierarchical trees and on the segmentation method of selecting the contextual region-of-interest (CROI) mask. The main idea is that all pixels corresponding to the same region are grouped in trees. The authors also propose a 17-step algorithm designed to obtain the compressed images. The results were analyzed using CR, MSE, PSNR, and CoC metrics. Also, the obtained images were compared to other existing methods such as JPEG, JPEG2000, Scaling, Maxshift, Implicit, and EBCOT. Ansari continues their work in their next paper [38], where different selection methods for the contextual region of interest are presented: a mathematical approach, segmentation approach, interactive approach, and the generation of an ROI mask. The proposed contextual set partitioning in a hierarchical tree algorithm is applied after the CROI is separated from the background. Also, the proposed contextual set partitioning in hierarchical tree (CSPIHT) algorithms is extended to 20 steps. The authors achieved compressions at various ratios from 10:1 to 256:1 and bpp from 1.0 to 0.03125 for the proposed CSPIHT algorithm. For testing purposes, an 8 bit image of size 667 × 505 was considered. The compression performance parameters such as BPP, CR, MSE, PSNR, and CoC were calculated for the proposed CSPIHT algorithm. The average PSNR for the ROI area was 36.54 dB with a CoC of 0.983.

Hierarchical lossless image compression aims to improve accuracy, reduce bitrate, and extend compression efficiency to improve the image storage and transmission procedure. Because usually only a part of the image is of high interest, the authors of paper [39] (Sumalatha et al.) focus on maintaining the quality of a defined contextual region. In their work, the ROI is encoded with AMWT (adaptive multiwavelet transform) using MLZC (multi-dimensional layered zero coding). As preprocessing, filtering (spatial adaptive mask filter) is applied to reduce noise. Then, the ROI is extracted and encoded using contextual multidimensional layered zero coding with high bits per pixel and a low compression ratio. The RONI is encoded using low bits per pixel and a high compression ratio. Finally, the ROI and the background are merged. This paper used the adaptive lifting scheme to derive the AMWT-filter coefficients. The predictor in the adaptive lifting scheme was adjusted to compute the current pixel using two prior values. The computational complexity is decreased by the suggested predictor. Experiments and tests were performed on an 8 bit image, with the usual metrics (PSNR, RMSE, CR, MAE, CC and MSSIM) analyzed. Also, a comparison between the proposed method and existing methods was performed. The compression ratio ranges from 1.23 up to 17.23 with PSNR values of 29.99 dB up to 43.25 dB.

The authors (Devadoss et al.) of paper [40] emphasize an image-compression model using block BWT-MTF with Huffman encoding alongside hybrid fractal encoding. The same approach for defining a region of interest is used in this paper too. Therefore, the critical zone is separated from the non-critical regions, and then, for the ROI, block-based Burrows–Wheeler compression is used. The rest of the image is encoded using a hybrid fractal encoding technique. The output image is reconstructed by merging the two zones. Burrows–Wheeler transform (BWT) groups similar repetitive elements by sorting the input data, therefore, making the compression more efficient. The move-to-front transform (MTF) has the purpose to transforming the local context to a global context by assigning indexes to symbols (the ones that occur more frequently have a smaller index). The splitting of the ROI from the other unimportant data is completed using morphological segmentation. As a first step of this process, the input image is turned into a grayscale image. Following that, structuring elements are assigned. Using morphological operators such as erosion and dilation, the input image is integrated with the specified structuring element. In the resulting binary image, pixels with a value of one are replaced with the original pixel value from the input image, resulting in the ROI component of the image. To obtain the NROI component, the binary image is inverted, and then the pixel with a value equal to 0 in the input image is replaced with the original pixel value. Because the compression algorithm has a high computational complexity, the ROI is divided into smaller blocks (128 × 128, 64 × 64, 32 × 32, 16 × 16, and 8 × 8). The performance of the proposed algorithms were evaluated using CR, space saving, time consumption, and PSNR quality metrics. The obtained values were compared to conventional methods. The average value for PSNR was 34.42 dB and for CR it was 11.67; multiple tests were conducted with different bpp values.

Another example of how the output of an ROI and RONI separation algorithm shall work is presented in Figure 3.

In the medical field, region-of-interest-based coding techniques have become more important due to their ability to compress and transmit data efficiently. Pre-processing images is the first step proposed by Kaur et al. in paper [41]. The image is then separated into ROI and non-ROI segments using segmentation. Lastly, compression is used to lower network and storage bandwidth. In this paper, two efficient compression methods—fractal lossy compression for non-ROI images and context tree weighting lossless for the ROI portion of an image—are proposed and compared with other methods, including scalable RBC and integer wavelet transform. When compared to earlier techniques, such as IWT and Scalable RBC, the suggested strategies have shown a low mean-squared-error rate, high PSNR, and high compression ratio. According to the results, the average CR is 89.6, and the average PSNR is 55.2 dB.

Fahrni et al., in [42], developed a novel approach to medical-image compression that offers the optimal trade-off between compression efficiency and image quality. The approach is predicated on various contexts and ROIs, which are determined by the level of clinical interest. Primary ROIs, or high-priority areas, are given a lossless compression. The background and secondary ROIs are compressed with moderate to severe losses. The primary region of interest (PROI), secondary region of interest (SROI), and background (NROI or RONI) are the three regions into which the authors suggest dividing the images. The primary region of interest in an image is called the PROI. This region can be defined by the radiologist manually or automatically using computer-aided segmentation (CAS) [43]. Since the PROI holds the most important information, it is best to keep this area intact. It will go through lossless compression, which will result in a small size reduction but will guarantee that the exam’s most important data are fully retained. Regions of lesser clinical interest that can be compressed with a moderate loss of visual quality are included in the SROI. The remainder of the image, which is of no real significance, is included in the background. Two methods were proposed one with a variable bit rate (MVAR) and one with a fixed bit rate (MFIX). The MVAR method obtained PSNR values over 40 dB and an average MOS of 5 (excellent). When executing the tests, the DICOM format was used to store the images, with a bit depth of 16 bpp and an in-plane resolution of 512 × 512 pixels. With a 9:1 compression ratio when compared to the original, non-compressed images, the average-compressed-image size was up to 61% smaller than that of standard compression techniques such as JPEG2000. The suggested approach also has the benefit of not requiring the conventional compression engine. Stated differently, the method can be applied to any ROI-based compression scheme. The idea of a multi-level compression approach could still be used in the future, with automatic segmentation of any pre-defined PROIs, SROIs, and background, to achieve even greater results in terms of compression ratio, while maintaining satisfactory image quality. This is especially true given the ongoing development of artificial intelligence (AI)-based methods [44].

### 4.2. Image Watermarking in Medical Field

There are domains in which the images contain sensitive diagnoses information that should be protected from any tampering risk. Therefore, an authentication method that assures the image security but does not affect its quality was introduced in parallel to the compression process. With this purpose, Badshah et al., in their paper [45], focus on the lossless compression of an image’s ROI and on the importance of watermarking using different techniques. The Lempel–Ziv–Welsh (LZW) technique is the one that stands out, with a compression ratio of 13, which is better than PNG, GIF, JPG, and JPEG2000. The LZW technique can preserve the ROI qualities, unchanged, while ensuring there is less payload encapsulation into the image. The LZW approach is based on a dictionary approach to achieve lossless compression that can be applied for both images and texts. An initial image with a size of 480 × 640 was taken and an ROI of 100 × 200 pixels was defined using image segmentation. Generally, the image’s central part can be defined as ROI as it is a more informative zone. Then, the ROI is used as a watermark in the watermarking process of images. The choice of a larger ROI can increase the LZW-image-compression ratio. ROI was obtained for the experiment phase and then transformed into binary values of zeroes and ones. Each binary-converted pixel value was added to a file. Binary values are repeated in sequences in the binary file meaning that greater size ROIs have more binary sequences that repeat, and higher sequence counts enhance the image’s compression ratio. The authors saw the benefits of LZW-based watermarking and developed a more robust approach in their next paper [46].

The work presented in [46] focuses on the watermarking of images to authenticate them and detect illegal changes. The process of digital watermarking refers to the actions taken to embed relevant information to an image with the purpose of copyright protection, recovery, and authentication. But, to maintain the image quality, the watermark shall be losslessly compressed, with the focus on Lempel–Ziv–Welch (LZW) losslessly compressed watermarks. The LZW method is a dictionary-based compression technique that can be used for text data and images. In this case, the watermark can be considered a combination of the defined region of interest and the applied secret key. In each image, a 100 × 100 pixel segment was selected as the ROI, and then a secret key was generated and applied to obtain the watermark. The watermark was processed with LZW lossless compression and placed in the image RONI portion. The compression ratio achieved using the LZW algorithm is compared to other utilized formats available, obtaining the best results with an average CR of 18.5. The average PSNR value for the watermarked image is around 54 dB. If LZW is used more than once, the entire watermark size can theoretically be reduced to two binaries, a zero and a one. The addition of a two bit watermark to an image will permanently fix the image deterioration and watermark accommodation issues. To verify the authenticity of the ROI area, the watermarking secret key that was used before the watermarking process is compared to the secret key obtained after performing the decompression. If the codes do not match then tamper localization and lossless recovery are required. Similarly, if both values match, it indicates that the ROI is genuine, and the image can be used for further research.

Watermarking an image is a common process because some sensitive information needs to be protected against illegal alterations. Haddad et al. [47] present a watermarking scheme based on the lossless compression standard JPEG-LS. The novelty is that the access to security services can be granted without decompressing the image. The JPEG-LS standard refers to a lossless or near-lossless compression, with the purpose of providing a low-complexity image-compressing algorithm proposed by the International Standards Organization ISO/IEC JTC1 [48]. The technique relies on the LOCO-I algorithm (lOw cOmplexity Lossless cOmpression for Images) [49] which is based on pixel prediction using a contextual statistical model. The values for the PSNR metric are greater than 40 dB (up to 50.5 dB), while the smallest capacity achieved regarding the bit of information per pixel of an image is 0.03. The visual-saliency-based index (VSI) is between 0.94 and 0.99, depending on the capacity (bit of information per pixel of image). Even if some information from the image was lost, the visual quality of the watermarked region is almost similar to the original. The experiments were conducted on a dataset consisting of 60 images with a pixel size of 576 × 690, with 8 bits per pixel. 

To precisely recover tampered with medical images, Liew et al. in [50] proposed the tamper-localization and lossless-recovery (TALLOR) scheme. This method allows the tampered image to be restored to its original state through lossless compression. The image that was recovered can be considered the same as the original and could still be utilized for a diagnosis. The problem with this plan was that the original image was compressed using lossless compression and then embedded as part of the watermark. As a result, it took longer to locate the tamper and recover the tampered image by decompressing and using the embedded watermark. The tamper localization and recovery processing would take longer if the user had asked for the image to be authenticated at the time of usage. In the next paper [51], the authors suggested improvements to their earlier work to decrease the recovery time and tamper localization by proposing a reversible watermarking scheme (TALLOR) by dividing the image into the ROI and RONI. The ROI is the important portion of medical images that physicians use to make diagnoses, and RONI is the region outside of the ROI. Using Jasni’s scheme [52], watermarking is carried out in the ROI region for tamper detection and recovery. After compression, the original least-significant digits (LSBs) that are eliminated during the watermark-embedding process are kept in RONI. The watermarking scheme can be reversible because the saved LSBs can be used later to return the image to its original bit value. By further segmenting the ROI into smaller parts, tamper localization and lossless recovery with ROI segmentation (TALLOR-RS) is an enhanced watermarking scheme. Each segment requires separate authentication. Only segments that are suspected of tampering can be further examined as part of a multilevel authentication process.

Tampering with sensible images can be prevented by applying watermarking. Usually, an image can be divided into two separate zones, the region of interest (ROI) and region of non-interest (RONI). The authors of paper [53] continued the work from [51], revealing a ROI-based tamper detection and watermarking that embeds the important information into the least-significant bits of the image which can be used anytime for data authentication and recovery. The watermarking process is described as a 14-step algorithm. The image can be split into multiple regions, each marked if they are of interest or not. The authors choose to define one ROI and five RONI. Then, the watermarking authentication process consists of a configuration step followed by the authentication process, verification process (where image is checked if it is corrupted or not), a process that checks that no tampering has occurred in the RONI areas where they were used to store ROI watermark bits, and, finally, the recovery of the image. Experiments were conducted to prove the usability and the performance of the shown work. The ROI-DR (detection and recovery) method was also compared to other existing methods. Because TALLOR and TALLOR-RS, the proposed watermarking schemes, share certain characteristics, like storing ROI bits in RONI’s LSB, implementing JPEG compression techniques, and using the SHA-256 hashing method in the algorithm, a comparison of these three watermarking schemes will be carried out, and the ROI-DR watermarking scheme’s speed-up factors in comparison to TALLOR and TALLOR-RS will be measured. These three watermarking schemes will conduct their experiments using the same hardware and software environment, as well as the same set of image samples for testing purposes, to ensure fairness in the comparison of results. The process of watermarking embedding proved to be faster than other existing schemes. The outcome of the experiment demonstrated that the ROI-DR is robust against different types of tampering and can restore the tampered ROI to its original form. It also achieved a good result in imperceptibility with peak signal-to-noise ratio (PSNR) values of roughly 48 dB. The images used for testing purposes had a size of 640 × 480 and 8 bits per pixel.

Tackling the subject of watermarking an image without affecting the quality of the output, the authors of paper [54] (Zermi et al.) proposed an innovative approach based on DWT (discrete wavelet transform) decompression. Singular-value decomposition is then applied on the three sub-bands (LL, LH and HL) which allows for retention of the maximum energy of the image. The sub-bands are combined to perform the watermark integration. The proposed method is divided into three parts: the generation of the watermark, the insertion process, and then the extraction of the watermark. Nevertheless, the watermark is not encrypted in the suggested method. An encryption can be added prior to the integration process to strengthen the security of the watermark. One of the advantages of the proposed work is its robustness to compression operations. Different metrics such as PSNR, SSIM, or NPCE were used to measure the performances obtained and to analyze the image distortions caused by the watermarking process. Experiments were made to check the output provided by the proposed approach. The average PSNR value was 53.54 dB, and the average SSIM was 0.9988.

### 4.3. Methods Used in Automotive

When discussing the automotive field, the development towards autonomous vehicles is clear; increasingly, well-known manufacturers have started developing smart vehicles that can handle certain situations on their own, with the driver only being required for supervision. In this context, the use of onvehicle cameras will increase, capturing images to be analyzed and processed to serve as an input for several security features. Akutsu et al. proposed in paper [55] an ROI-based image-compression algorithm that is meant to be used inside an infrastructure-quality-control system. The purpose of this system is to use the cameras available on a vehicle to detect the structural quality of the road; this means that a large amount of data needs to be transmitted and stored for further processing, making image compression a vital part of the whole system. The proposed compression method uses annotation information such as bounding boxes or segmentation maps for a defined weighted quality metric that is based on SSIM. This will be used for the region of interest part, with the convolutional auto encoder (CAE) being used for the non-important regions. Experiments were conducted on a dataset containing damaged-road images of 256 × 256 pixels alongside annotation data. The results show that the method can be compared to other existing methods (BPG—better portable graphics). Even if the PSNR for the ROI is slightly lower (27 dB compared to 33 dB for BPG), the bits per pixels were reduced by 31%, meaning that the quality is almost preserved, but the size is considerably reduced.

The interest in autonomous vehicles has led to the need for wireless communication between the elements involved in traffic (vehicle to vehicle, vehicle to infrastructure, infrastructure to vehicle communication). Realizing that the increase in cameras and sensors involved in vehicle to X communications will lead to hard-to-reach processing requirements and bottlenecks. Löhdefink et al. [56] proposed an ROI-based image-compression algorithm that focuses on the reduction of overall data size. The ROI is defined using a binary mask which uses semantic segmentation networks to extract important information. The network was trained using a loss function applied only on the ROI. Therefore, the sender acquires the image, and then semantic segmentation and the ROI mask generator are applied before the encoding of the data. For testing purposes, a dataset containing almost 4000 training, validation, and testing images was used. The size of each picture was downscaled to 1024 × 512 pixels due to limited GPU video memory. The PSNR for the ROI was around 27 dB. As seen, for now, the neural network approaches do not weld the same results as other methods, but for certain scenarios they can be useful, especially for selecting the ROI.

In time, the vehicles will rely increasingly on artificial vision, meaning that the large amount of data that must be transmitted will become a problem, especially knowing that, historically, in the automotive field, the available resources are rigorously checked and limited to reduce the costs of production. ROI-based image-compression algorithms can be very useful for infrastructure-only applications such as speed traps where only the license plate can be considered important. Efficient image compression can be used in systems that check whether the road tax was paid or not; in this scenario, the license plate is again a region of interest with the other ROI being a sticker on the windshield that is usually the proof of payment. Another example where a compression method could be useful is the application described in [57] where the visibility on a portion of a road during foggy weather is estimated using a laser, LIDAR, and cameras. 

### 4.4. Methods Used for Spatial Images and Remote-Sensing Images

Hyperspectral sensors have gained a lot of popularity recently for remote sensing on Earth. These systems can provide the user with images that contain information about both the spectrum and the spatial domain. Present-day hyperspectral spaceborne sensors have improved spectral and spatial resolution, enabling them to capture large areas. Because of their limited storage capacity, it is necessary to lower the volume of data acquired on board to prevent a low orbital duty cycle. The focus of the recent literature has been on effective methods for on-board data compression. The harsh environment (outer space) and the constrained time, power, and computing resources make this a difficult task. Graphic processing unit (GPU) hardware characteristics have frequently been used to speed up processing through parallel computing. In the presented work [58] (which continues the work from [59]), a GPU on-board operating framework is proposed by Giordano et al., utilizing NVIDIA’s CUDA (Compute Unified Device Architecture) architecture. The algorithm uses the related strategy of the target to perform on-board compression. Specifically, the primary functions involve using an unsupervised classifier to automatically identify land cover types or detect events in nearly real time in regions of interest (this is a user-related option). This means, compressing regions using space-variant different bit rates, such as principal component analysis (PCA), wavelet, or arithmetic coding and managing the volume of data sent to the ground station. One impediment can be that supervised classification cannot be used because the labels, in this case, the topographic classes, are unknown in advance; unsupervised classification techniques are better suited for the automatic segmentation and identification of ROIs in hyperspectral images. Algorithms for clustering detect meaningful patterns without label knowledge and do not require training stages. The distortion index (DI), which is the inverse of the mean square error (MSE) between the reconstructed and original image, has been used to gauge accuracy. Rather than using the entire image (ROI and background), the DI has only been calculated in the ROI area to provide a fair comparison. The algorithm has superior performance in the ROI part compared to JPEG2000.

In his work, Zhang et al. [60] proposed a deep-learning self-encoder framework-based region-of-interest (ROI) compression algorithm to enhance image-reconstruction performance and minimize ROI distortion. Since most traditional ROI-based image-compression algorithms rely on manual ROI labeling to achieve region separation in images, the authors first adopted a remote-sensing image cloud detection algorithm for detecting important targets in images. This involves first separating the important regions in the remote-sensing images from the remote-sensing background and then identifying the target regions. To synthesize images and to more effectively reduce spatial redundancy, a multiscale ROI self-coding network from coarse to fine with a hierarchical super priority layer was designed. This significantly improved the distortion-rate performance of image compression. The authors improved compression performance by employing a spatial-attention mechanism for the ROI in the image-compression network. Using an accurate key information enhancement mechanism, the suggested algorithm can compress the ROI more successfully and enhance the impact of image compression using a higher bit rate. The Landsat-8 satellite-image dataset was used to train and test the designed ROI image-compression algorithm. Its performance was then compared to that of the conventional image-compression algorithm and with deep-learning image-compression algorithms without the attention mechanism. Two different methods were tested: coarse to fine hyper prior modeling (CFHPM) for learned image compression, and multiscale region of interest coarse to fine hyper prior modeling (MROI-CFHPM) for learned image compression, with the second one providing better results with a PSNR value of up to 37.5 dB. MROI-CFHPM can be used to achieve the differential compression of the ROI, significantly enhancing the overall image’s compression performance.

Focusing on the fact that wireless image-sensor networks are usually poorly equipped nodes, with a camera, a radio transceiver, and a limited processor, Kouadria et al., proposed in paper [61], a solution that reduces energy consumption for the image-compression process. Using discrete Tchebichef transform (DTT) as an alternative to the discrete cosine transform (DCT), the ROI of the image will be compressed, with the expected outcome being that less computational complexity is required, and, therefore, less energy is consumed. The goal is to achieve the same compression ratio as other state-of-the-art methods but with a reduced number of operations. The approach taken is to have an image as a reference frame and a separate sensor that is used for movement detection. When that sensor provides a trigger, a new frame will be taken; the ROI will be generated based on the differences between the new frame and the reference image. Therefore, when transmitting, only the region of interest will be compressed using DTT and forwarded; the rest of the image will be discarded. The key element in the proposed method is the change-detection algorithm. Different datasets containing pedestrians or cars moving on roads have been used (with different sizes: 240 × 320, 768 × 576 or 512 × 512 pixels). The obtained performances for the PSNR metric are around 40 dB, which can be considered good enough, but the main advantage of ROI-only compression based on change detection and DTT is the fact that the energy consumption was reduced by half compared to classical formats such as JPEG. The only drawback of the method is that the testing was conducted using a simulation, not a real-life scenario.

## 5. Results and Discussions

Categorization of the ROI-based compression techniques’ performances is necessary for anyone (individuals or companies) that is interested in the image-compression subject. Different performance metrics will be taken into consideration, with the aim being to present the lowest value of that metric, the highest, and the average, but only if enough information is available. The main performance metrics that will be analyzed for ROI-based techniques but also for other compression methods are the compression ratio (CR), peak signal-to-noise ratio (PSNR), and mean-square error (MSE). When discussing the quality-related metrics, there are many variables that must be taken into consideration. Each author used different test images with different sizes and characteristics, and different regions of interest were defined for each technique. This is why the metrics were presented as an interval (minimal and maximal values) and were available so that the reader can gather an idea of what the expected performances are of a certain algorithm used. If the results were presented for multiple images, we would also compute the average value, which can provide an indication of the expected performance. The field of use of each image is very important because the input will differ greatly based on the area in which it is used.

In Table 1, the different hybrid methods are presented from the compression-ratio (CR) point of view. The higher the CR value the more space is saved when an image is stored, and the more time is saved when an image is transmitted, but also the quality can be affected. The compression ratio of the whole image is of interest in this case, and it is usually dependent on the size of the region of interest.

Table 2 highlights the PSNR metric. A higher value for the PSNR metric represents a better image quality, and it is typically used to test the considerable distortion introduced by the compression and decompression processes between the input and output image. Therefore, an ideal compression technique would yield a high compression ratio and a high PSNR. A value of over 40 dB can be considered excellent. The PSNR of the ROI part is of interest since the RONI is considered to have low-value information.

Table 3 shows the registered MSE values which can provide a perspective on the difference between the input and output images. A lower value of MSE means a more efficient compression algorithm, since there are less differences between the compared images. The MSE of the ROI part is of interest since the RONI is considered to have low-value information.

To determine if a real advantage is gained from the ROI-based approaches, Table 4 highlights the CR performances of whole-image-compression techniques. Any lossless or lossy compression technique that compress the whole image without focusing on a region of interest can be considered as a general compression technique. For these approaches the same alghorithm is applied for the whole image.

In Table 5 the PSNR values are presented for analyzed whole-image-compression techniques.

Table 6 presents the performances of whole-image-compression techniques from the MSE metric point of view.

In all presented tables, also, the field in which the techniques were tested is mentioned. Even if each method is designed for a specific domain, its applicability is not restricted to a single use case.

As the established name says, regions of interest, are the areas of an image that must be retrieved (extracted) with the best-possible resolution and then transmitted or stored (safely and with minimum memory-space consumption) in the shortest possible time to be able to correctly reconstitute the selected areas (with minimal or no losses), analyze, and interpret or diagnose specific situations. These aspects are required not only for vehicles of autonomous driving, traffic safety systems, or telemedicine (MRI, CT, US) where the detection of diseases through images occurs in real time—where images are seen by several doctors in different locations—but also in other fields such as quality control of materials, chemical structures, biological structures, etc. 

In most applications, the ROIs are of interest. If the non-ROI part is also necessary, then the user must decide how it is more advantageous to segment and compress the image. Usually, the non-ROI also contains some low-value information, so different compression algorithms could be used for ROI and non-ROI. In this case, the most common approach is to extract the ROI and leave black pixels [6,8] in the position where the ROI was. If the background is not important at all, a single compression algorithm can be used for the ROI, and the non-ROI can be discarded, replacing the whole background with black pixels [38,43,51], or even clipping [41,53] parts of the image if possible. When restoring the image (concatenating/merging the areas in which the ROI was tagged), the appropriate decisions are made for the purpose pursued. Artifacts or discontinuities can be created [39,42,46] when ROIs and non-ROIs are merged after decompression to restore the image depending on how well the segmentation, compression, decompression, and fusion methods used are. 

## 6. Conclusions

Image-compression technology is vital for the efficient and cost-effective long-term storage and archiving of image data. This is particularly relevant in fields where preserving historical or scientific images is a priority, for example in the medical field where the history of a patient may need to be saved and stored for decades. Watermarking is an important procedure to assure the authenticity of the received information. With compressed images, organizations can manage their archives more effectively, ensuring that valuable visual data remain accessible and intact. Efficient image compression has a positive environmental impact as by reducing the size of image files it lessens the energy consumption of data centers and reduces carbon emissions associated with data storage and transmission. This contribution to sustainability is increasingly important as the world grapples with environmental challenges. Looking at the future of autonomous driving, the images captured by cameras must be sent to a main processor and then analyzed in real-time; this means that the data transmission must be optimized. Apart from this, the storage capabilities are usually limited; therefore, the need for space saving is elemental.

The speed at which images load and the overall quality of visual content have a profound impact on the user experience. Websites and applications that load images quickly and smoothly are more likely to engage users and retain their attention. As a result, the performance of the chose image-compression algorithm has a direct influence on user satisfaction and interaction with digital platforms. Furthermore, the digital landscape is characterized by an array of devices with different screen sizes and resolutions. Image compression ensures that images are adaptable and compatible with a variety of platforms, ranging from smartphones and tablets to desktop computers. This versatility is indispensable for creators and developers seeking to reach a broad and diverse audience.

When compressing an image, almost every time a decisional point is reached where the appropriate balance between space saving and maintaining the image quality must be found. As in most aspects concerning technologies, there cannot only be benefits, drawbacks are inevitable. Therefore, if an image is compressed in a lossless manner, it will maintain its quality, but the space savings will be insignificant. The other variant would be applying a lossy compression which will result in high space savings, but the image quality will not be preserved. Trying to overcome the situation mentioned above, region-of-interest-based methods have become increasingly popular. The main idea of a ROI-based method being to define a region inside the image which will be preserved, yielding low space savings. The major reduction in size will come from the region of noninterest which does not contain important information, and which can be sacrificed from a quality point of view to gain a satisfactory compression ratio. The drawback is that a new processing step is introduced, and the selection of ROI brings extra computational cost. Nevertheless, ROI-based algorithms proved to be the right choice when only part of the image is of high interest and the other part cannot be removed. It is obvious that selecting a smaller ROI will result in a better overall compression ratio.

Traditional 2D compression algorithms, designed for processing two-dimensional data, handle each image slice independently. This independent processing is simpler and less computationally intensive. However, their lower computational complexity and faster processing times make them advantageous in scenarios where rapid image processing is essential, and the data does not exhibit significant inter-slice correlation like in live medical procedures. In contrast, 3D compression algorithms are tailored to handle volumetric data, considering the spatial correlations between different layers or slices. This approach typically results in higher compression ratios for 3D datasets, as these algorithms leverage redundancies across the volumetric data for more effective compression, all at a higher computational cost.

The JP3D algorithm [62] is specifically tailored for 3D image compression, providing efficient compression for volumetric data. It has become a popular choice in medical imaging due to its high compression ratios and preserved image quality. Similarly, 3D set partitioning in hierarchical trees (SPIHT), an extension of the 2D SPIHT algorithm, is renowned for its efficiency in compressing 3D medical images, providing excellent compression ratios, while maintaining high image quality. The adaptation of 3D wavelet transforms to medical imaging effectively reduces the size of volumetric medical images by capturing spatial and frequency information at different scales.

Choosing between 2D and 3D compression methods is based on specific medical-imaging requirements, balancing the need for real-time processing, image quality, and diagnostic efficacy. 

In this work, a study of different ROI-based techniques presented in the latest decade was conducted, the most important aspects of each work being highlighted alongside important quality metrics. This study was conducted with the aim of helping researchers to choose a specific compression method without conducting an in-depth search of the literature. The preferred methods and quality metrics that were found in the literature were presented. To reach a better idea of the real advantages over the conventional compression methods, Table 7 presents the average CR, PSNR, and MSE values for ROI-based and general compression methods. The presented values in Table 7 take into consideration all the papers presented in this article.

As can be seen, ROI-based approaches have better performances using all the considered metrics as algorithms that process the whole image using the same approaches. The general methods can achieve a threshold compression ratio and if that is exceeded the image quality will suffer greatly. Due to the main principles behind them, the contextual methods can overcome these limitations. The definition of what is important in an image and what is not can be either derived automatically or by hand; this part is vital for the whole compression process. After all, it is up to the user to determine if it makes sense to use an ROI-based algorithm, or not, depending on the specific use case. Due to the vast amount of image data, a major field in which ROI-based image compression is used is telemedicine and medical imaging.

The particularities of the compression algorithms for the regions of interest depend on the restrictions imposed by the certain application that uses those ROIs. In such situations, the non-ROI part is not of interest, and it can be decided to be discarded [53], or to be compressed with less performing algorithms (under various aspects: data losses, time consumption, hardware resources, etc.) [8,40,60].

From the authors point of view if a specific region of the image is of high interest and the rest of the image still has some low-priority information, an ROI-based approach would be the best choice, as it will still preserve the quality of the designated zone and will also result in major space reduction. The additional cost for the obtained benefits is the increase in the process complexity.

## Figures and Tables

**Figure 1 sensors-24-00791-f001:**
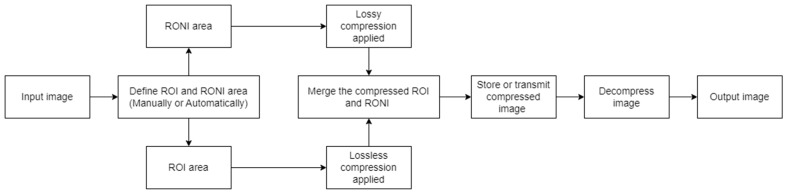
Region-of-interest-based-compression/decompression approach.

**Figure 2 sensors-24-00791-f002:**
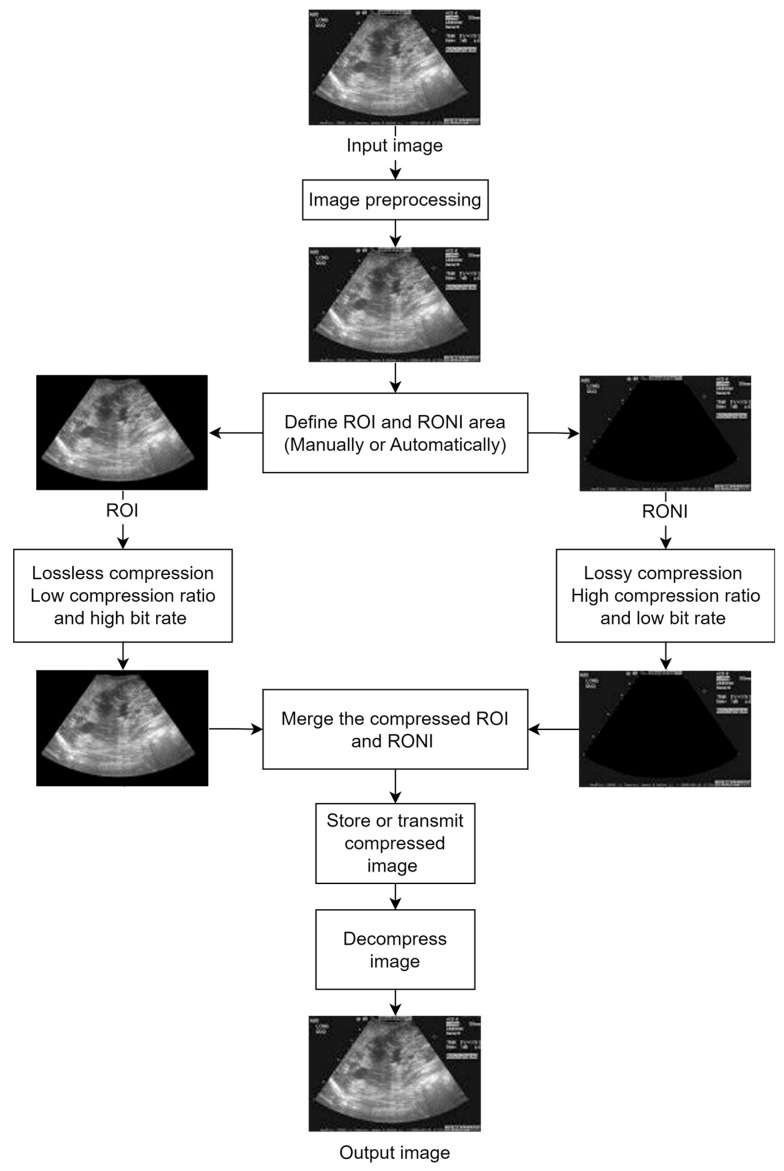
Example of how an ROI-based algorithm works; image based on [36].

**Figure 3 sensors-24-00791-f003:**
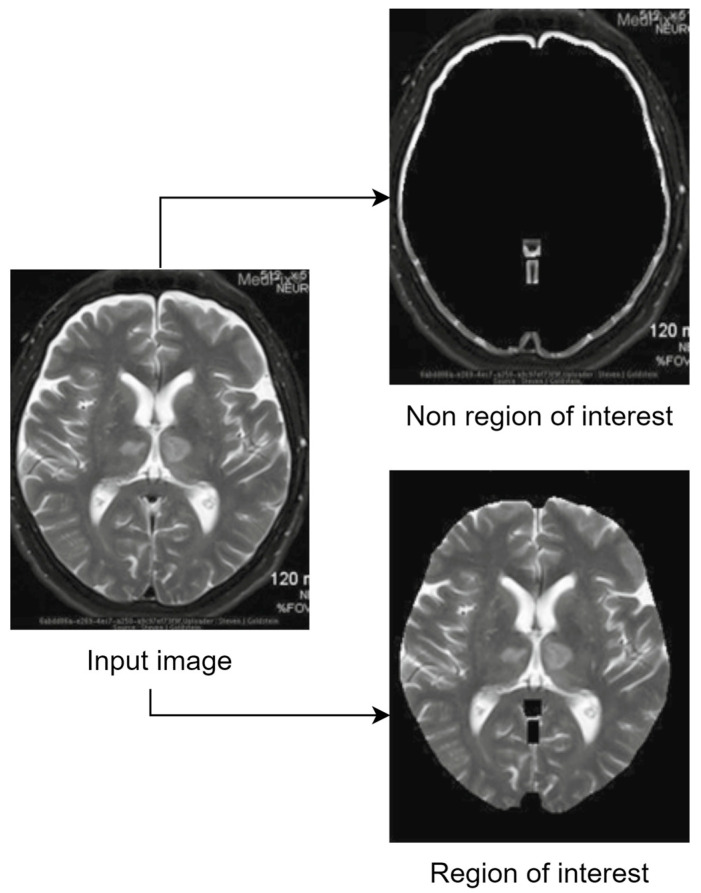
Example of separating the input image into ROI and RONI; image based on [40].

**Table 1 sensors-24-00791-t001:** Hybrid compression techniques are presented using the CR metric.

Paper Reference	CR Min	CR Max	CR Average	Field of Use
Hosseini et al. [36]	11	256	62.1	Medical, telemedicine-magnetic resonance imaging (MRI), X-ray, computer tomography (CT), ultrasound imaging (US)
Ansari et al. [37,38]	10.56	256.5	62.29	
Sumalatha et al. [39]	1.29	17.23	4.41	
Devadoss et al. [40]	4.94	22.26	11.67	
Kaur et al. [41]	85.71	93.12	89.6	
Fahrni et al. [42]	N.A.	N.A.	9	
Badshah et al. [45,46]	11.36	47.61	23.1	
Liew et al. [50,51,53]	1.38	2.77	1.66	

**Table 2 sensors-24-00791-t002:** Hybrid compression techniques presented after PSNR metric.

Paper Reference	PSNR Min (dB)	PSNR Max (dB)	PSNR Average (dB)	Field of Use
Hosseini et al. [36]	38.27	42.36	40.16	Medical, telemedicine-magnetic resonance imaging (MRI), X-ray, computer tomography (CT), ultrasound imaging (US)
Ansari et al. [37,38]	34.04	37.26	36.54	
Sumalatha et al. [39]	29.99	43.25	36.77	
Devadoss et al. [40]	30.22	39.53	34.42	
Kaur et al. [41]	49.01	60.1	53.27	
Fahrni et al. [42]	40.25	57.85	46.6	
Badshah et al. [45,46]	51.53	56.53	53.99	
Haddad et al. [47]	47.25	50.5	48.87	
Liew et al. [50,51,53]	48.17	49.59	48.94	
Zermi et al. [54]	44.92	57.04	53.54	
Akutsu et al. [55]	N.A.	N.A.	27.04	Automotive-Autonomous driving
Löhdefink et al. [56]	N.A.	N.A.	27.24	
Zhang et al. [60]	29.5	37.5	33.5	Spatial images and remote-sensing images
Kouadria et al. [61]	38	42	40	

**Table 3 sensors-24-00791-t003:** Hybrid compression techniques presented after MSE metric.

Paper Reference	MSE Min	MSE Max	MSE Average	Field of use
Hosseini et al. [36]	3.78	9.69	6.46	Medical, telemedicine-magnetic resonance imaging (MRI), X-ray, computer tomography (CT), ultrasound imaging (US)
Ansari et al. [37,38]	12.19	25.61	14.84	
Devadoss et al. [40]	5.88	28.78	21.7	
Kaur et al. [41]	0.15	2	1.03	
Fahrni et al. [42]	0.1	6.13	3.15	
Zermi et al. [54]	0.12	0.34	0.2	

**Table 4 sensors-24-00791-t004:** Classical compression techniques presented after CR metric.

Paper Reference	CR Min	CR Max	CR Average	Field of Use
Dokur et al. [20]	16.44	31	21.43	Medical, telemedicine-magnetic resonance im-aging (MRI), X-ray, computer tomography (CT), ultrasound imaging (US)
Viswanathan et al. [21]	2.48	15.18	5.6	
Vallathan et al. [22]	6.69	8.1	7.72	
Nemirovsky et al. [24]	10	52	31	
Ammah et al. [25]	N.A.	N.A.	91.25	
Shahhoseini et al. [26]	4.5	15	9.75	
Janet et al. [27]	10.76	15.53	14.18	
Krivenko et al. [28]	6	78	42	
Cheng et al. [30]	2.5	7.7	5.1	
Mansour et al. [31]	1.57	3.59	2.45	
Cao et al. [32]	5	35	20	Spatial images and remote sensing images
Wei et al. [34]	1.33	4	2	General–Color and greyscale known images
Wang et al. [35]	2	20	11	

**Table 5 sensors-24-00791-t005:** Classical compression techniques presented after PSNR metric.

Paper Reference	PSNR Min (dB)	PSNR Max (dB)	PSNR Average (dB)	Field of Use
Viswanathan et al. [21]	34.46	54/64	42.43	Medical, telemedicine-magnetic resonance imaging (MRI), X-ray, computer tomography (CT), ultrasound imaging (US)
Vallathan et al. [22]	31.36	44.73	37.63	
Nemirovsky et al. [24]	32	35.2	33.7	
Ammah et al. [25]	47.53	71.27	61.16	
Shahhoseini et al. [26]	26.8	27.8	27.3	
Janet et al. [27]	32.8	35.61	34.44	
Krivenko et al. [28]	37.5	52.5	45	
Cao et al. [32]	7.12	9.08	8.2	Spatial images and remote sensing images
Löhdefink et al. [33]	20.5	27.23	24.11	Automotive-autonomous driving
Wei et al. [34]	28.81	42.09	35.59	General–color and greyscale known images
Wang et al. [35]	21	40	30.5	

**Table 6 sensors-24-00791-t006:** Classical compression techniques presented after MSE metric.

Paper Reference	MSE Min	MSE Max	MSE Average	Field of Use
Dokur et al. [20]	116.67	185.9	147.49	Medical, telemedicine-magnetic resonance imaging (MRI), X-ray, computer tomography (CT), ultrasound imaging (US)
Vallathan et al. [22]	2.2	9.54	7.11	
Ammah et al. [25]	N.A.	N.A.	46.49	

**Table 7 sensors-24-00791-t007:** Hybrid methods compared to classical compression techniques.

Compression Techniques	Average CR	Average PSNR (dB)	Average MSE
Hybrid	32.97	41.49	7.89
Classical	20.26	34.55	67.03

## Data Availability

Data are contained within the article.

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
