# Peer review of "Image-Compression Techniques: Classical and “Region-of-Interest-Based” Approaches Presented in Recent Papers"

_sensors, 2024, doi:10.3390/s24030791_

Round 1

Reviewer 1 Report

Comments and Suggestions for Authors

the paper presents a comprehensive review of recent developments in ROI-based image compression techniques, highlighting their importance in various fields and their efficiency in preserving image quality while optimizing storage and transmission.

In all the tables in the paper, you need a horizontal border for the Field of Use column, because it's not clear the item is assigned to which row(s)

In the paper, sometimes CT is referred to as computed tomography sometimes it's computational time. Please clarify the specific meaning of CT.

Line 192: there is a typo: whare--> where 

Lots of the CT MRI images are 3D, please brief compare 3D compression algorithms with 2D algorithms

Comments on the Quality of English Language

NA

Author Response

Thank you for review and for the valuable suggestions that will help us improve our work. Below we have added a response to each point.

Point 1: In all the tables in the paper, you need a horizontal border for the Field of Use column, because it's not clear the item is assigned to which row(s).

R: Very good point to increase readability. All tables have been updated as suggested to increase clarity.

Point 2: In the paper, sometimes CT is referred to as computed tomography sometimes it's computational time. Please clarify the specific meaning of CT.

R: Computational Time is also called Running Time therefore RT abbreviation will be used. The CT abbreviation will only refer to Computed Tomography. The lines 185 – 188 were updated in the updated version of the manuscript:

“Computational Time (also called running time RT) is defined as the amount of time required in the process of compression and decompression of an image. An efficient technique must have less computational time. The RT is usually obtained using different time measurement tools and techniques.”

Point 3: Line 192: there is a typo: whare--> where 

R: Typo corrected. Authors performed an English check of the whole manuscript.

Point 4: Lots of the CT MRI images are 3D, please brief compare 3D compression algorithms with 2D algorithms.

R: The following paragraphs were added inside the updated manuscript (lines 994 - 1014):

“Traditional 2D compression algorithms, designed for processing two-dimensional data, handle each image slice independently. This independent processing is simpler and less computationally intensive. However, their lower computational complexity and faster processing times make them advantageous in scenarios where rapid image processing is essential, and the data does not exhibit significant inter-slice correlation like in live medical procedures. In contrast, 3D compression algorithms are tailored to handle volumetric data, considering the spatial correlations between different layers or slices. This approach typically results in higher compression ratios for 3D datasets, as these algorithms leverage redundancies across the volumetric data for more effective compression all at a higher computational cost.

The JP3D algorithm is specifically tailored for 3D image compression, providing efficient compression for volumetric data. It has become a popular choice in medical imaging due to its high compression ratios and preserved image quality. Similarly, 3D Set Partitioning in Hierarchical Trees (SPIHT), an extension of the 2D SPIHT algorithm, is renowned for its efficiency in compressing 3D medical images, providing excellent com-pression ratios while maintaining high image quality. The adaptation of 3D Wavelet Transforms to medical imaging effectively reduces the size of volumetric medical images by capturing spatial and frequency information at different scales.

Choosing between 2D and 3D compression methods is based on specific medical imaging requirements, balancing the need for real-time processing, image quality, and diagnostic efficacy.”

Reviewer 2 Report

Comments and Suggestions for Authors

This manuscript is a review of comparative research on detection and compression techniques for regions of interest in images over the last decade.  The paper cites a large amount of literature, which is rich in content, including an introduction to the various metrics needed to measure image compression based on ROI and the whole image , but it needs to be further improved if it is to be published in this journal. The

main problems are shown as follows:

1. In the main manuscript, every paragraph of the paper describes the methods and innovations of a certain paper. But only basically describing the methods and experimental results of the relevant papers is not enough. Some researching status, existing problems and possible future development trends can be added before and after the paragraphs to make the connection of the article more coherent and generalized. For example, starting with line 687, "a lossy technique designed on wavelet transform to compress images and to preserve the clinical The paragraph "information" does not have much to do with "An alghorithm for lossless compression of ultrasound sensor data" from line 687 in the previous paragraph, and some transfer relations can be added.

2. The sections numbers are confused. "General image compression approaches" is Chapter 4, the "Results and discussions" of the next chapter should be Chapter 5 instead of Chapter 4, and the sequence number of the next chapter also needs to be modified. Moreover, I suggest the section of general image compression locate before the section ROI-based approaches. 

3. The abbreviation "MROI-CFHPM" appears for the first time in line 586, while the term "multiscale region of interest coarse to fine hyper prior modeling" already appears in line 584. You should add "(MROI-CFHPM)" after the first occurrence of the word, and the abbreviation can be directly referenced in the following text.

4. It is better to classify, compare and analyze the related references to make them more clear for readers. The experiments like Table 2. ROI-based compression techniques compared after PSNR metric. should explain wether under the same CR, if not, it is meaningless. The other results have the same problems.

Author Response

Thank you for review and for the valuable suggestions that will help us improve our work. Below we have added a response to each point.

Point 1: In the main manuscript, every paragraph of the paper describes the methods and innovations of a certain paper. But only basically describing the methods and experimental results of the relevant papers is not enough. Some researching status, existing problems and possible future development trends can be added before and after the paragraphs to make the connection of the article more coherent and generalized. For example, starting with line 687, "a lossy technique designed on wavelet transform to compress images and to preserve the clinical”. The paragraph "information" does not have much to do with "An alghorithm for lossless compression of ultrasound sensor data" from line 687 in the previous paragraph, and some transfer relations can be added.

R: We agree with this point and did our best to improve our content. Transfers from one paragraph to other were improved, alongside the grouping of paragraphs so papers from the same domain are place one after another. Extra details regarding possible problems, in which environment the tests were executed, or possible ways of improvement were added. Some paragraphs were reformulated for better cohesion. For example, paragraph from line 678 (in the old manuscript) was moved in a more suited section of the chapter in the updated manuscript (lines 386 - 415). Another example is the restructuration of lines 417 – 447 and 359 - 385 to better describe the results and the performances, alongside some drawbacks. Several small changes that help the readability were performed in other paragraphs too.

Point 2: The sections numbers are confused. "General image compression approaches" is Chapter 4, the "Results and discussions" of the next chapter should be Chapter 5 instead of Chapter 4, and the sequence number of the next chapter also needs to be modified. Moreover, I suggest the section of general image compression locate before the section ROI-based approaches. 

R: Indeed, a better content organization was required. The chapters order was updated, and the numbering sequence was corrected. Chapters were renamed to increase clarity. The new structure is:

  1. Classical image compression approaches

3.1 Methods used in medical imaging and telemedicine

3.2. Methods used for spatial images and remote sensing images

3.3. Methods used in automotive

3.4. Methods not targeting a specific domain

  1. Hybrid image compression approaches

4.1. Methods used in medical imaging and telemedicine

4.2. Image watermarking in medical field

4.3. Methods used in automotive

4.4. Methods used for spatial images and remote sensing images

Chapter 4 “Hybrid image compression approaches” was extended with sub-chapter 4.3. “Methods used in automotive” and sub-chapter 4.4. “Methods used for spatial images and remote sensing images” was extended with one more paper.

Point 3: The abbreviation "MROI-CFHPM" appears for the first time in line 586, while the term "multiscale region of interest coarse to fine hyper prior modeling" already appears in line 584. You should add "(MROI-CFHPM)" after the first occurrence of the word, and the abbreviation can be directly referenced in the following text.

R: "(MROI-CFHPM)" was added after the first occurrence of the word as suggested. Lines 871 – 876:

“Two different methods were tested: coarse to fine hyper prior modeling (CFHPM) for learned image compression and multiscale region of interest coarse to fine hyper prior modeling (MROI-CFHPM) for learned image compression, the second one proving better results with a PSNR values up to 37.5 dB. MROI-CFHPM can be used to achieve the differential compression of the ROI, significantly enhancing the overall image's compression performance.”

Point 4: It is better to classify, compare and analyze the related references to make them more clear for readers. The experiments like Table 2. ROI-based compression techniques compared after PSNR metric. should explain whether under the same CR, if not, it is meaningless. The other results have the same problems.

R: The paper aims to be a study on the techniques and performance indicators used in recent works and not a comparison of the performance indicators in compression for each image of each analyzed work. Unfortunately realizing research in which different compression techniques are applied on the same input image even if it can be useful, it is almost impossible to achieve. It would require contacting each authors/journal for usage wrights on their images, datasets, and algorithms (which sometimes can be very difficult to obtain). We tried to better express our intentions in chapter 5 and 6.   

Chapter 5 “Results and discussions” was updated on lines 903-913:

“The main performance metrics that will be analyzed for ROI-based techniques but also for other compression methods are compression ratio (CR), Peak Signal to Noise Ratio (PSNR) and Mean Square Error (MSE). When discussing about the quality related metrics there are many variables that must be taken into consideration. Each author used different test images with different sizes, characteristics, and different regions of interest were defined for each technique. This is why the metrics were presented as an interval (minimal and maximal values) were available so the reader can get an idea of what are the expected performances if a certain alghorithm is used. If the results were presented for multiple images, we also computed the average value which can give an indication of the expected performances. A very important information is related to the field of use of a certain image because the input will differ greatly based on the area where it is used.”

Chapter 6 “Conclusions” was updated on lines 1015-1023:

“In this work, a study of different ROI-based techniques presented in the latest decade was conducted, the most important aspects of each work being highlighted alongside important quality metrics. This study was done with the aim of helping researchers to choose a specific compression method without doing an in-depth literature search. The preferred methods and quality metrics that were found in literature were presented. To get a better idea of the real advantages over the conventional compression methods, Table 7 presents the average CR, PSNR and MSE values for ROI-based and general compression methods. The presented values in Table 7 take into consideration all the papers presented in this article.”

Reviewer 3 Report

Comments and Suggestions for Authors

This manuscript provides a review mainly focusing ROI (region of interest)-based techniques in the field of imaging compression through a comparative study based on some primary performance metrics such as CR (compression rate), PSNR (peak signal to noise ratio) and MSE (mean square error). The analysis results may be interesting, though not unexpected deep-going, and will have some helpful to readers who want to seek more effective imaging compression methods.

Some suggestions and problem may help authors to improve this manuscript before being accept publishing.

(a) Among the section “3. ROI-based approaches”,it would be better if some little titles can be abstracted according to the technical contents in those literatures, to present the technological developing in this field, this, meanwhile, would add readable for these materials.   

(b) There repeats the numbering between the section ‘4. General image compression approaches’ and the section ‘4. Results and discussions’.

(c) It would be much better if it could be a big reduce in word number from the paragraph 1 to the paragraph 5 which are mostly redundant expressing on the importance and meaningful of imaging compression, and this manuscript just aims the effective compression.

(d) Lines 123-125, ‘Therefore, the idea of non-ROI approaches is to use lossless compression to maintain every single component of a medical image.’ Why?

(e) Lines 136-137, ‘Because an image's compression ratio (CR) is inversely related to its ROI, precisely locating the diagnostic zone from medical pictures becomes difficult.’ Why?

Comments on the Quality of English Language

The writing may be improved, as above mentioned. 

Author Response

Thank you for review and for the valuable suggestions that will help us improve our work. Below we have added a response to each point.

Point 1: Among the section “3. ROI-based approaches”,it would be better if some little titles can be abstracted according to the technical contents in those literatures, to present the technological developing in this field, this, meanwhile, would add readable for these materials.  

Point 2: There repeats the numbering between the section ‘4. General image compression approaches’ and the section ‘4. Results and discussions’. 

R: We treated both points in one major update. We agree that a better content structure was necessary. Subtitles based on the technical content of the literature were added for chapters 3 and 4.  The chapters order was updated, and the numbering sequence was corrected. Chapters were renamed to increase clarity. The new structure is:

  1. Classical image compression approaches

3.1 Methods used in medical imaging and telemedicine

3.2. Methods used for spatial images and remote sensing images

3.3. Methods used in automotive

3.4. Methods not targeting a specific domain

  1. Hybrid image compression approaches

4.1. Methods used in medical imaging and telemedicine

4.2. Image watermarking in medical field

4.3. Methods used in automotive

4.4. Methods used for spatial images and remote sensing images

Chapter 4 “Hybrid image compression approaches” was extended with sub-chapter 4.3. “Methods used in automotive” and sub-chapter 4.4. “Methods used for spatial images and remote sensing images” was extended with one more paper.

Point 3: It would be much better if it could be a big reduce in word number from the paragraph 1 to the paragraph 5 which are mostly redundant expressing on the importance and meaningful of imaging compression, and this manuscript just aims the effective compression.

R: We tried to remove unnecessary or repeated information. The size of the first 5 paragraphs was reduced to remove redundancy, 2 paragraphs being removed. Redundant paragraphs removed (from the old manuscript) from lines 55 - 66 and lines 77 – 89.

Point 4: Lines 123-125, ‘Therefore, the idea of non-ROI approaches is to use lossless compression to maintain every single component of a medical image.’ Why?

R: We agree that the text was not clear enough. The paragraph was reformulated to improve the clarity and readability. Lines 97 – 110 were updated:

“In domains such as medical image processing, the large size of images, the need of efficient storage and the critical requirement of preserving the details and the clarity of the images (so the experts can provide an accurate diagnosis based on them), created the need of a hybrid compression method [5]. Therefore, two areas have been defined, region of interest (ROI) and region of not interest (RONI or non-ROI). Important diagnostic information is contained in medical photographs, which must be retained during compression. The classical compression approaches apply the same lossless compression alghorithm to the whole image to maintain every single component of a medical image. Preserving diagnostic features is the reason behind using lossless compression technology on medical photographs. However, because the regions of high importance are treated with the same priority and in the same manner as the background of the image, the resulting compression ratio (CR) will be low and the transmission of the image on the web requires a high bandwidth [6]. The ROI-based compression technique can be used to alleviate this issue [7].”

Point 5: Lines 136-137, ‘Because an image's compression ratio (CR) is inversely related to its ROI, precisely locating the diagnostic zone from medical pictures becomes difficult.’ Why?

R: We agree we didn’t express what we wanted to say with enough clarity. The paragraph was reformulated to improve the clarity and readability. Lines 111 – 133 were updated:

“The term "region of interest" (ROI) refers to the area of a picture that is more significant than other areas [8], while non-ROI refers to the rest of the image. Appling lossy compression on non-ROI and lossless compression on ROI leads to maintain the necessary diagnostic features of the medical image while achieving a high compression ratio. Utilizing ROI-based compression has the benefit of offering a high compression ratio without compromising the quality of a significant portion of the image [9]. Because an image's compression ratio is strongly related to its ROI size (a smaller ROI means better CR), precisely defining the size and the location of the diagnostic zone from medical pictures becomes very important. It is critical to appropriately detect ROI since it includes crucial diagnostic information that can’t be altered, and the CR of the image is dependent on its size. The detection of ROI is a vital task since incorrectly located ROI might result in the loss of diagnostic data in medical pictures. One of the first steps in any ROI-based compression technique is ROI selection. The complexity and execution duration of any approach are determined by how ROI is chosen. Some common ROI selection strategies (such as region growth and saliency maps) are employed frequently by different researchers, or sometimes the ROI is chosen manually if the images follow a certain pattern. After the regions are defined, different compression techniques are applied to preserve the region of interest and to save space by sacrificing the non-ROI. Usually, to reconstruct the input image, the inverted process is applied during the decompression step. Figure 1 shows the general process of ROI-based compression and highlights the main steps that are followed to achieve hybrid image compression. Of course, that extra steps such as filtering, salt and pepper noise reduction or image watermarking (used for authentication and tampering prevention) are to be expected.”

Reviewer 4 Report

Comments and Suggestions for Authors

Low novelty

Author Response

Point 1: Low novelty

R: Thank you for your review. We noticed that there is no study in the recent years that focuses on the region of interest-based image compression algorithms and decided to fill that void. Therefore, this study was done with the aim of helping researchers to choose a specific compression method without doing an in-depth literature search. The preferred methods and quality metrics that were found in literature were presented in our work. We tried to highlight in which area the methods were used, how the performances were measured, on what images the tests were done and what are the values of some quality metrics.

On our other studies we often relied on a review paper as a starting point for our research, this is the reason we believed that the present review has value, and it can be useful for other researchers.